# Airways Type-2 Related Disorders: Multiorgan, Systemic or Syndemic Disease?

**DOI:** 10.3390/ijms25020730

**Published:** 2024-01-05

**Authors:** Francesco Giombi, Gian Marco Pace, Francesca Pirola, Michele Cerasuolo, Fabio Ferreli, Giuseppe Mercante, Giuseppe Spriano, Giorgio Walter Canonica, Enrico Heffler, Sebastian Ferri, Francesca Puggioni, Giovanni Paoletti, Luca Malvezzi

**Affiliations:** 1Otorhinolaryngology Head & Neck Surgery Unit, IRCCS Humanitas Research Hospital, Via Manzoni 56, Rozzano, 20089 Milan, Italy; francesco.giombi@gmail.com (F.G.); francesca.pirola@humanitas.it (F.P.); michele.cerasuolo@humanitas.it (M.C.); fabio.ferreli@hunimed.eu (F.F.); giuseppe.mercante@hunimed.eu (G.M.); giuseppe.spriano@hunimed.eu (G.S.); luca.malvezzi@humanitas.it (L.M.); 2Department of Biomedical Sciences, Humanitas University, Via Rita Levi Montalcini 4, Pieve Emanuele, 20090 Milan, Italy; giorgio_walter.canonica@hunimed.eu (G.W.C.); enrico.heffler@hunimed.eu (E.H.); sebastian.ferri@humanitas.it (S.F.); francesca.puggioni@hunimed.eu (F.P.); giovanni.paoletti@hunimed.eu (G.P.); 3Otorhinolaryngology Head & Neck Surgery Unit, Casa di Cura Humanitas San Pio X, Via Francesco Nava 31, 20159 Milan, Italy; 4Personalized Medicine, Asthma and Allergy, IRCCS Humanitas Research Hospital, Via Manzoni 56, Rozzano, 20089 Milan, Italy

**Keywords:** chronic rhinosinusitis, asthma, personalized medicine, atopic dermatitis, type-2 inflammation

## Abstract

Chronic rhinosinusitis (CRS) has recently undergone a significant paradigm shift, moving from a phenotypical classification towards an “endotype-based” definition that places more emphasis on clinical and therapeutic aspects. Similar to other airway diseases, like asthma, most cases of CRS in developed countries exhibit a dysregulated type-2 immune response and related cytokines. Consequently, the traditional distinction between upper and lower airways has been replaced by a “united airway” perspective. Additionally, type-2 related disorders extend beyond respiratory boundaries, encompassing conditions beyond the airways, such as atopic dermatitis. This necessitates a multidisciplinary approach. Moreover, consideration of possible systemic implications is crucial, particularly in relation to sleep-related breathing diseases like Obstructive Sleep Apnoea Syndrome (OSAS) and the alteration of systemic inflammatory mediators such as nitric oxide. The trends in epidemiological, economic, and social burden are progressively increasing worldwide, indicating syndemic characteristics. In light of these insights, this narrative review aims to present the latest evidence on respiratory type-2 related disorders, with a specific focus on CRS while promoting a comprehensive perspective on the “united airways”. It also introduces a novel concept: viewing these conditions as a multiorgan, systemic, and syndemic disease.

## 1. Introduction

The most recent guidelines by the European Position Paper on Rhinosinusitis and Nasal Polyps (EPOS), defined chronic rhinosinusitis (CRS) as a long-lasting (>12 weeks) inflammation of the nasal cavity and paranasal sinuses, presenting symptoms of nasal obstruction/congestion or nasal discharge, possibly associated with facial pain/pressure or a dysregulated sense of smell [1]. In EPOS2020, CRS has been classified according to etiology, anatomic distribution, endotype, and phenotypes, but the main clinical subgroups of CRS are distinguished by endoscopic findings in a variant with nasal polyps (CRSwNP), and another without nasal polyps (CRSsNP). Based on the disease’s phenotype, taking into account different endoscopic, radiographic and clinical aspects, several clinical entities have been further proposed in years, including central compartment atopic disease (CCAD), allergic fungal rhinosinusitis (AFRS), or aspirin-exacerbated respiratory disease (AERD) [1]. While their definitions may be clear, several aspects of these conditions are still far from being fully comprehended, both at the molecular and clinical level. Indeed, CRS is a generic term that may be useful for establishing a diagnosis, but it is widely incomplete in defining the full complexity of different clinical patterns.

In recent years, due to the progress in scientific research, we have come to recognize CRS not merely as a localized disease but as an immunological alteration with local and systemic manifestations. Consequently, the concept of inflammatory endotype, particularly type 2 inflammation in our case, has been introduced in this field [2]. Therefore, the recognition of the pathobiological mechanisms driving the disease’s development has led to a paradigm shift. This shift involves moving away from a “phenotype-based” definition, in order to embrace an “endotype-based” approach, which has gained significant importance in EPOS2020 [1,3,4].

“Health” was defined in 1948 by the World Health Organization as a state of complete physical, mental, emotional, and social well-being and not only the absence of disease or infirmity [5]. Consequently, the entire healthcare system is progressively recognizing the importance of approaching inflammatory diseases of the upper airways (UA) with a “patient-centered” perspective rather than a “disease-centered” one. This shift places primary emphasis on the subjective well-being of the patient rather than solely on the objective absence of disease [6].

In order to improve the effectiveness of this novel personalized approach, any information directly provided by the patient concerning a health condition and its management (Patient Reported Outcomes—PROs) represents a fundamental component of any treatment paradigm [7]. The degree or severity of symptoms can be assessed and graded using many different tools: in a Likert-type scale as severe, moderate, mild, or asymptomatic, or as a visual analogue scale (VAS) score giving a measurable continuum (e.g., 0–10 cm) [8,9]. The assessment of PROs also allowed for the comparison between patients differently treated for similar conditions, thus becoming a critical quantitative measure for evaluating the efficacy of different therapies [10].

Despite the availability of diverse medical and surgical approaches for addressing and treating CRS, the burden of the disease remains significant on epidemiological, economic, and social fronts. Furthermore, the often interconnected nature of CRS with other UA diseases, whether viewed from a pathophysiological or clinical standpoint, has the potential to mutually amplify the burden of both conditions [11]; hence, the distinction between upper and lower airways is artificial, since the upper and lower respiratory tracts are both components of one, “united airway” system [12,13,14].

Moreover, adopting a “united airway” perspective extends beyond asthma to encompass various chronic pulmonary conditions. Additionally, the widely accepted concept of the rhino-bronchial syndrome emphasizes the interconnected pathophysiology between the upper and lower airways. Bearing in mind the concept of “united airways” from both an anatomical and pathophysiological perspective, in this review we aim to shine a broad spotlight on respiratory type-2 related inflammatory disorders, with particular attention to CRS, and propose a novel concept of such diseases as a multiorgan, systemic and syndemic condition.

## 2. A Multiorgan Condition

The classification of CRS in CRSw/sNP, based solely on the presence of nasal polyps (NPs), despite being still frequently used, does not provide an appropriate understanding of the disease in order to propose adequate treatment options [15].

To overcome the outdated “phenotype-driven” in favor of an “endotype-driven” classification of CRS, it is imperative to examine each patient comprehensively, considering clinical, endoscopic, radiological, and immunological perspectives [16]. Indeed, several immunological mechanisms that underlie CRS, such as type-2 inflammation, play a pivotal role in various other conditions, often of allergic nature. This influence impacts lower airway conditions like extrinsic asthma, but it also can extend broader than the respiratory apparatus, reaching other organs such as the skin and gastrointestinal tract. This feature profiles the condition as a multiorgan disease.

### 2.1. Type-2 Inflammation: Pathophysiology and Nasal Implications

In Western countries, the molecular mechanism most frequently responsible of primary CRS is the so-called “type-2 inflammation” [16]. However, the prevalence of type-2 related CRSwNP has been increasing also in Asia over the past decades [17]. Type-2 immune responses are characterized by the secretion of specific interleukins (Ils) such as IL-4, IL-5, IL-9 and IL-13. In detail, type-2 immunity induces a complex response involving granulocytes (eosinophils, basophils), mastocytes, type-2 innate lymphoid cells (ILC2), IL-4- and/or IL-13-conditioned macrophages and T helper 2 (Th2) cells [18]. These cells play a crucial role In the pathogenesis of CRS and related disorders. Consequently, understanding the mechanisms that regulate the intensity, maintenance, and resolution of type-2 immunity is crucial for comprehending disease progression and it is essential for therapeutic purposes. Under physiological conditions, these mediators serve a protective function by binding to extracellular pathogens. However, when dysregulated, their activity may become pathogenic [19].

One potential theory that seeks to explain the rising prevalence of type-2 related conditions in developed countries is the “hygiene hypothesis”, initially introduced by Strachan in 1989. This hypothesis originated from Strachan’s observation of an inverse correlation between hay fever and the number of older siblings while studying more than 17,000 British children born in 1958 [20]. According to this theory, the decreasing incidence of infections sustained by extracellular pathogens in Western countries and, more recently, in developing countries would be the origin of the disruption of the delicate equilibrium between type-1 (IL-2, IL-12, TNF-α, IFN-γ, lymphotoxin-α) and type-2 (IL-4, IL-5, IL-9, IL-13) inflammation [21]. Consequently, the reduction of external type-1 stimuli (e.g., bacterial and viral infections) would be compensated by a proportional increase of type-2 related immunity. Type-2 induced interleukins, although being phylogenetically directed against helminths, when dysregulated may promote the cross-reaction of immunological reactivity against self-epitopes, thus increasing the incidence of both autoimmune and allergic diseases [22]. Due to several advances in research, the Strachan’s hypothesis has been criticized and readapted over the years [23,24,25]. Epidemiologic studies provided evidence that several viral pathogens, although inducing a Th-1 related immune response, were not protective against allergic diseases, and, in many cases, even increased the risk [26]. Indeed, viral-induced damage to the epithelial barrier results in overexpression of Th-2 related cytokines such as IL-33, IL-25, and Thymic Stromal Lymphopoietin (TSLP), eventually exacerbating atopy [27]. Furthermore, a relevant contribution to the tolerogenic immune status has been assigned to the microbiota and to its complex interplay with the host’s mucosal surface. Recent studies have showed a higher taxonomic diversity in the gut microbiome of populations who maintained a primitive close-to-nature lifestyle compared to those used to permanently living in urban settings [28]. High-fiber diet and, in particular inulin, plays an effective role in enhancing the growth of *Bifidobacterium* and *Lactobacillus* species, which promote a tolerogenic status by releasing anti-inflammatory molecules such as TGF-β and IL-25. Accordingly, McLoughlin et al. applied soluble inulin to asthmatics in a 7 days placebo-controlled-trial, observing a significantly reduced number of eosinophils in the sputum as well as an improved global asthma control in patients orally treated with inulin [29]. Traveling from the gastrointestinal to the respiratory tract, the microbiota established in the lung might directly contribute to the pathogenesis of respiratory atopic disorders, from a perspective of a “gut-lung axis” [30]. A pivotal moment in the complex ontogenesis of a tolerogenic immune status is held during the pre- and neonatal period. Commensal acidophil bacteria start colonizing the human gut from as early as the fetal period, yet they continue during the passage through the birth canal and during breastfeeding [31]. Current evidence suggests that dysregulation of the neonatal microbiome may also contribute to exacerbate the Th-2 related axis. Accordingly, a recent analysis by the DIABIMMUNE study group found a marked drop in microbial gut diversity in Finnish and Estonian children, who instead hosted several *Bacteroides* species, eventually turning into an increased susceptibility to autoimmune processes, such as type-1 diabetes [32]. From this perspective, the term “hygiene hypothesis” is simplistic and misleading, since it fails to consider other factors now linked to the increase in type-2 related disorders, namely the dysregulation of tolerogenic axis, driven by the gut microbiome.

In the context of CRS, it seems that damaged airway epithelium initiates and perpetuates type-2 inflammation, and there is increasing evidence that environmental factors may enhance type-2 immunological response by stimulating epithelial expression of Th-2 related cytokines (e.g., TSLP, IL-25 and IL-33).

TSLP triggers dendritic cell-mediated type-2 inflammatory responses and enhances type-2 cytokine production in mast cells [33]. In support of its role in CRSwNP pathogenesis, the concentration of TSLP mRNA as well as its activity were found to be significantly higher in NPs tissue from patients with CRSwNP compared with uncinate tissue from patients with CRS or control subjects [34]. The stimulation of TSLP promotes an Intense cellular response by activating mast cells, ILC2 and releasing IL-5, IL-13, and IL-4. The product of this immune cascade is the activation of eosinophils (by means of IL-5 and IL-13) and B-cells (through IL-4) [35], resulting in proliferation of Th2 cells, isotype class switching of B cells to produce IgE, and eventually inducing airway hyperreactivity, mucus hyperproduction, as well as smooth muscle proliferation and fibrosis (e.g., tissue remodeling). Both IL-4 and IL-13 induce the abovementioned conditioned state of macrophages, a typical feature of CRSwNP [36,37].

One more molecule that is crucial for the pathogenesis or airway type-2 related disorders is the transforming growth factor-β (TGF-β) [38,39]. TGF-β is a key factor in the remodeling process found in sinonasal mucosa with CRS; specifically, TGF-β pathways were found to be upregulated in CRSsNP and downregulated in CRSwNP [40]. TGF-β upregulation leads to proliferation of fibroblasts, increased collagen deposition and extracellular matrix (ECM) production, hence resulting in fibrosis and basement membrane thickening [41]. In CRSwNP, TGF-β downregulation contributes to degradation of ECM and deposition of albumin, which results in intense edematous stroma, subepithelial and perivascular inflammatory cells infiltration, formation of pseudocysts and polypoid degeneration. As a result of the abovementioned mechanisms, TGF-β was also shown to be involved in smooth muscle remodeling, which is a typical feature of bronchial asthma [42].

Current evidence indicates that determinants of disease development extend beyond the dysregulated immune response. As abovementioned, other factors such as the microbiome and genetic predisposition play a pivotal role [43,44]. In addition, exposure to extrinsic agents is also recognized as a possible risk factor; however, although it has been established that air pollution interacts directly with airway mucosa, yet little is known about how pollutants affect UA inflammation. Many pollutants, including particulate matter (PM) and ozone (O^3^) have been shown to upregulate reactive oxygen species, leading to DNA damage and increased oxidative stress and inflammation [45,46].

The Impact of air pollution on CRS pathogenesis, severity, and progression has been recently systematically reviewed by Leland et al., who showed pollutant exposure, particularly PM, to be associated with higher odds of developing CRS. Increased air pollution was even associated with worsened disease severity and detectable histopathologic changes, such as eosinophilic aggregates, and Charcot–Leyden crystals in patients with CRSwNP [47,48].

### 2.2. The Lower Airways: Bronchial Asthma

Asthma is a condition of acute, fully reversible airway inflammation, often following exposure to an environmental trigger. The pathological process begins with the inhalation of an irritant (e.g., cold air) or an allergen (e.g., pollen), which then, due to bronchial hypersensitivity, leads to airway inflammation and to an increase in mucus production, creating an obstructive reversible bronchial condition [49,50].

The way asthma has been defined and classified has changed over time. Even If It was initially considered as “a single entity” disease, the individuation of the etiopathological mechanisms behind this condition allowed a novel definition of a “type-2 related” (or “Th2-high”) and a “type-2 unrelated” (or “Th2-low”) asthma [51]. It has been ascertained that type-2 is probably the most common type of asthma, since about 50–70% of asthmatics display an underlying type-2 inflammation pattern, as measured through eosinophils and IgE blood count [52]. From a clinical point of view, type-2 related asthma generally has a more severe symptomatologic burden, even though it is often more responsive to inhalant glucocorticoids and to inhibitors of type-2 inflammation.

In a recent real-life analysis, severe asthma, based on the European Respiratory Society/American Thoracic Society guidelines (ERS/ATS) [53], demonstrated a frequent association with several type-2 related pathways, including higher blood and sputum eosinophils, indicating that treatments targeting the most clinically severe forms of asthma should specifically include type-2 tailoring molecules [54].

At a bronchial level, IL-4 and IL-13 act by mediating inflammatory and remodeling changes in the airway, thus predisposing to the development of disease. Even if the functional roles of IL-4 and IL-13 are quite overlapping, it is, however, plausible that these two cytokines exert distinct pathobiological actions in asthma. In fact, IL-4 is the key inducer of CD4+ T-helper cell commitment towards a Th2 immunophenotype, whilst IL-13 primarily promotes the development of bronchial inflammation and remodeling, thus enhancing airway hyperresponsiveness [55]. IL-13 is involved in goblet-cell hyperplasia and mucus production, as well as smooth muscle contractility and hyperplasia. Moreover, IL-13-mediated damage to the epithelial barriers is also associated with the development of mucus plugs as a consequence of mucus hyperproduction [56]. IL-5 and IL-13 both play a role in B-cell class switching and IgE production, leading to the degranulation of eosinophils and mast cells and subsequent release of pro-inflammatory mediators as well as barrier disruption and tissue remodeling [57].

The aforementioned mediators induce tissutal changes such as smooth muscle hypertrophy, goblet cell metaplasia and hyperplasia, and subepithelial fibrosis which are at the basis of pathogenesis of bronchial asthma. Eosinophils can further exacerbate airway remodeling due to their release of TGF-β and cytokines by interactions with mast cells, worsening inflammation and aggravating asthma over time if not managed timely [58]. In a prospective cohort of 1267 asthmatics enrolled by Schatz et al., patients with well-controlled asthma at baseline were significantly more likely to maintain well-controlled asthma over the following year (76.2–80.4%), rather than patients with uncontrolled asthma at baseline (33.5–36.9%; *p*-value < 0.001) [59].

In addition, the alteration of structural elements induced by dysregulated type-2 immune patterns may contribute to an exaggerated response to inhaled antigens which, in a narrowed susceptible airway, predisposes to asthmatic exacerbations. Indeed, genetic population studies showed that gain-of-function single nucleotide polymorphisms of the IL-4 receptor gene IL4Rα are associated with severe asthma exacerbations, lower lung function, and increased mast cell-related tissue inflammation [60].

In support of the “united airways” perspective, data from 695 patients from the SANI register (Severe Asthma Network in Italy) have been recently analyzed by Canonica et al., who calculated a 40.6% prevalence of CRSwNP in this cohort. Similarly, atopic dermatitis and FeNO values (a reliable marker of the severity of inflammation of the lower airways) were significantly more frequent in patients with CRSwNP than in subjects without NPs. Finally, patients with CRSwNP had a significantly higher number of asthma exacerbations per year, greater intake of oral corticosteroids (OCS) and were more likely to be systemic corticosteroids users on the long term [61].

Further data supporting a role for type-2 inflammation in bronchial asthma pathogenesis and exacerbations come indirectly from clinical trials, including specific inhibitors of these particular pathways, such as omalizumab (Xolair^®^, Genentech USA, Inc., San Francisco, CA, USA and Novartis Pharmaceuticals Corporation, Tokyo, Japan) a recombinant humanized anti-IgE monoclonal antibody and others specific for type-2 cytokines (IL-4, IL-5, and IL-13) [62,63]. EXTRA was a phase IIIb multicentric, randomized, double-blind, placebo-controlled study which assessed the effectiveness of omalizumab in subjects with moderate to severe persistent asthma who are inadequately controlled with high-dose inhaled corticosteroids and long-acting beta-agonists. During 48 weeks of treatment, the rate of protocol-defined asthma exacerbations was significantly reduced for omalizumab compared with placebo, representing a 25% relative reduction [64]. Furthermore, at the end of the observation period, omalizumab improved the mean AQLQ score (Asthma Quality of Life Questionnaire, a standardized PRO for asthmatic patients), reduced mean daily albuterol puffs and decreased the total asthma symptom severity score, which included a nocturnal asthma score, morning asthma symptoms, and a daytime asthma symptom score [65].

In addition, two drugs that act by blocking circulating IL-5 have been developed: mepolizumab (Nucala^®^, GSK London, England, UK) and reslizumab (Cinqaero/Cinqair ^®^, Teva Pharmaceutical Industries, Tel Aviv, Israel). In a phase III trial (MENSA). Mepolizumab was evaluated at two doses (75 mg intravenous or 100 mg subcutaneous) in 576 subjects aged 12 to 82 years with a baseline peripheral eosinophil count ≥ 150 cells/μL and ≥2 exacerbations in the year prior despite the use of a high dose of inhalant corticosteroids. Over a 32-week treatment phase and 8-week follow-up safety phase, subcutaneous (SC) and intravenous (IV) mepolizumab reduced annualized exacerbation rates by 47% and 53%, respectively, relative to placebo. Moreover, in patients with a peripheral eosinophil count ≥ 500 cells/μL, mepolizumab reduced exacerbation rates by 74 and 79% relative to placebo with SC and IV doses, respectively [66].

Therefore, the efficacy of Th2 inflammation modulating target drugs in enhancing severe uncontrolled asthma cases should be regarded as supplementary evidence of the pathophysiological mechanism of the disease and its close association with the type-2 immune response and related cytokines.

### 2.3. Other than the Airways: Atopic Dermatitis

In the context of a dysregulated type-2 immune response, the dermis is one more anatomical site where the pathological consequences of the abovementioned biological pathways are more clearly evident. Atopic dermatitis (AD), also known as atopic eczema, is a long-lasting inflammatory disease of the dermis, resulting in itchy, red, swollen, and cracked skin [67]. The epidermis plays a crucial role in the pathogenesis of the disease, since it acts as a physical and functional barrier, and skin surface defects are the most significant pathologic findings in AD [68]. Filaggrin (FLG), transglutaminases, keratins, and intercellular proteins are key factors responsible for epidermal function and downregulation of these molecules facilitates allergen and microbial penetration into the skin. Accordingly, skin barrier dysfunction has been considered the first step in the onset of the “atopic march” as well as AD. In this context, however, type-2 inflammation plays a dual role, both promoting the Initiation of skin defect as well as the maintenance of the inflammatory process [69]. The upregulation of IL-4 and IL-13 has been demonstrated to inhibit the expression of epidermal proteins, such as FLG, loricrin, and involucrin, leading to skin barrier defects [70]. In addition, elevated levels of IL-13 mRNA have been detected in both lesional and non-lesional skin of AD patients, and the number of IL-13-producing circulating T cells has been closely associated with disease severity [71,72]. Furthermore, recent studies have suggested that TSLP produced by keratinocytes may serve as a trigger, activating dendritic cells to secrete chemokines, which attract Th2 cells to the skin, finally releasing proallergic cytokines (e.g., IL-4, IL-5, and IL-13) [73]. Accordingly, the upregulated expression of TSLP has been reported in the skin of AD patients, thus producing a vicious circle where skin barrier disruption induces type 2 inflammation and type 2 inflammation increases barrier disruption.

Several randomized controlled trials assessed the efficacy of targeted type-2 immune response inhibitors in AD, which, again, indirectly confirms the role of this pathobiological mechanism in the development of disease. Lebrikizumab is a high-affinity IgG4 monoclonal antibody targeting interleukin-13, therefore preventing the formation of the IL4Rα-IL13Rα1 heterodimer receptor signaling complex. Its effectiveness in AD was recently proved in two 52-week, randomized, double-blind, placebo-controlled, phase 3 trials, whose primary outcome was achieving a reliable clinical improvement in terms of the Investigator Global Assessment score (which measures the degree of involvement of the skin). In trial #1, the outcome was met in 43.1% of 283 patients in the lebrikizumab group and in 12.7% of 141 patients in the placebo group (*p*-value < 0.001), whereas in 33.2% of 281 patients in the lebrikizumab group and in 10.8% of 146 patients in the placebo group in trial #2 (*p*-value < 0.001) [74].

At present, several evidence have demonstrated the etiopathogenesis of many other clinical conditions to be correlated with dysregulated type-2 immune response, at different anatomic subsites. At a gastrointestinal level, as it is for eosinophilic esophagitis, as well as at an ophthalmological level, in case of allergic conjunctivitis. It should certainly be considered that type-2 related cytokines are unlikely to be the main trigger factor of all these conditions, which are often driven by specific pathophysiological features (e.g., airway hyperreactivity and bronchial constriction for asthma, nasal mucosa swelling/congestion for CRS, skin epithelial barrier dysfunction for AD), but rather a shared mechanism which simultaneously predisposes and amplifies the host’s pathological response to external stimuli. Because of the different specific features within a dysregulated type-2 immune response, tailored therapy may be directed towards key pathways other than type-2 related. For instance, the activation of Notch2 by ligation with Jag1 and subsequent nuclear translocation of the Notch2 intracellular domain has been observed to induce bronchial epithelial dysfunction by increasing the differentiation of club cells into goblet cells rather than ciliated cells, hence promoting mucus plugs formation and airway obstruction [75]. Accordingly, antisense oligonucleotide-mediated inhibition of Jag1/Notch2 pathway has shown promising results in murine models and could provide a novel therapeutic path for the treatment of multiple chronic respiratory diseases [76]. Kallikrein-7, a serine protease involved in several homeostatic processes, has been speculated to be a significant contributor to AD pathogenesis, since it was found overexpressed in lesional AD skin. Likewise, loss-of-function mutations in SPINK5 resulted in upregulated epidermal Kallikrein activity, resulting in AD-like symptom in both mice and humans [77].

Considering these novel insights, different pathways will likely be targetable in specific diseases in the future; however, type-2 inflammation still represents a common thread between these conditions. The detailed discussion of any type-2 related disorder and its specific triggers goes beyond the intentions of this review; however, it should give a contribution to switch the perspective towards a more holistic approach, addressing type-2 related conditions from a multidisciplinary and personalized perspective.

## 3. A Systemic Condition

The growing understanding of shared pathophysiological features among UA conditions, their frequent coexistence, and the anatomical continuity between the upper and lower airways has prompted a shift from the disease-centered “one-size-fits-all” concept to a more comprehensive and personalized approach. This approach considers individual differences in genes, environment and lifestyles, adopting a “united airways” perspective. In this scenario, it is recommended to assemble a multidisciplinary team comprising, at the very least, otolaryngologists, allergologists/clinical immunologists, pneumologists, and psychologists. This collaborative approach aims to provide the most effective diagnostic and therapeutic strategies for the patient, especially in an era where emerging therapeutic interventions, such as biologic agents, are shaping the landscape of managing type-2 related disorders of the airways [78].

In 2006, Swarbrick presented a “wellness approach” by comparing it to the traditional medical model. This method, subsequently endorsed by esteemed associations such as the American Psychological Association (APA), encompassed eight different but mutually co-dependent dimensions of wellness: emotional, occupational, social, spiritual, intellectual, environmental, financial, and physical [79].

The multidisciplinary approach should consider the possible further Implications of airways type-2 related disorders in the different dimensions of wellness. According to the “wellness approach” by Swarbrick, the definition of the physical sphere comprehends all areas of health that relate to the physical aspects of the body including exercise, weight management, ergonomics, disease prevention and sleep quality. Eventually, airways type-2 related disorders show further implications on several physical spheres, mainly sleep quality, hence take on the features of a systemic syndrome.

### 3.1. Further Implications: The Link with Sleep-Related Respiratory Disorders

In recent years, interest in the relationship between airways type-2 related diseases and sleep-related breathing disorders has been raising, mainly due to the increasing epidemiological trend and socioeconomic burden of both these conditions.

Obstructive sleep apnea (OSA) is characterized by recurrent episodes of UA occlusion during sleep (apnoeic or hypopneic events) with consequent excessive daytime sleepiness and/or cardiovascular dysfunctions, hence the clinical scenario of OSA syndrome (OSAS) is currently widely accepted [80].

The complex pathophysiology of OSA is still far from being fully comprehended. Between the multitude of etiopathogenetic mechanisms, however, nasal obstruction plays a crucial role, since it has been demonstrated to produce a switch to oral night breathing. Although this mechanism bypasses the nasal airflow blockage, oral breathing during sleep is physiologically disadvantageous, since it promotes a decrease in the retroglossal diameter, leading to an increased UA resistance and pharyngeal collapse, thus possibly resulting in snoring and increased apnea/hypopnea episodes [81].

A recent study in 810 patients of the Icelandic Sleep Apnea Cohort showed that nocturnal nasal congestion occurs at least once per week in 65% of untreated OSA patients and at least three times a week in 35% of patients. Moreover, patients with nasal obstruction reported more daytime sleepiness (Epworth sleepiness scale score 12.5 ± 4.9 vs. 10.8 ± 5.0; *p*-value < 0.001) [82].

In a real-life analysis by Friedman et al., patients with mild OSA showed a worsening of polysomnographic parameters (apnea-hypopnea index, oxygen desaturation index and snoring) after iatrogenic nasal obstruction (e.g., post-surgical packaging), possibly due to exacerbating the principal obstruction at the level of the nasal airway. On the other hand, no significant results were observed for patients with severe OSA, who are most likely to present a multilevel obstruction, where the nasal airway is only a cofactor of a more heterogeneous condition [83]. Finally, a positive correlation between nasal obstruction (as measured by acoustic rhinometry) and the Respiratory Disturbance Index (which includes apneic, hypopneic episodes as well as respiratory efforts-associated arousals) has been demonstrated in nonobese patients [84].

Apart from nasal obstruction, several inflammatory mediators, including type-2 related cytokines such as IL-4, have also been investigated as having possible effects on sleep regulation, thus unravelling the codependence of those two apparently different clinical conditions. In a comparative study by Krause et al., levels of pro-allergic serum cytokines in allergic subjects correlated with increased latency to rapid eye movement sleep, decreased time in rapid eye movement sleep, and decreased latency to sleep onset [85].

Cytokines released by muscle fibers may initiate the activation and migration of inflammatory cells into affected areas of the muscle. The inflammation within the UA of OSA patients has the potential to induce myocontractile dysfunction and weakness of dilator muscles through at least two distinct but interrelated mechanisms. First, inflammatory cell infiltration, together with production of proinflammatory mediators, such as cytokines and oxygen free radicals, can significantly weaken muscle fibers, promoting the airway collapse and sleep-related obstruction and oxygen desaturation. Moreover, some inflammatory mediators (IL-6, TNF-α) may induce a damage to the efferent nervous fibers, resulting in a weaker stimulation of the pharyngeal dilatator muscles [86].

Accordingly, Boyd et al. demonstrated a significant increase in inflammatory T-cell infiltration of the UA in patients with OSA, which encompassed both the mucosal and muscular layers [87]. Moreover, muscle changes mirroring both myopathy and neuropathy have been shown coexisting in biopsies from the soft palate of OSA patients [88].

An Increasing concern for sleep-related breathing disorders has been raised due to the knowledge of its increasing epidemiological burden as well as its impact on the whole healthcare process. The prevalence of primary/simple snoring in middle-aged men has been reported in the range of 25–50%, with several differences between different geographical areas [89,90], whereas recent epidemiological studies (such as the recent “HypnoLaus”) have shown that the prevalence of moderate-to-severe OSA, also due to the increased sensitivity of modern diagnostic facilities, stands up to 23.4% in women and 49.7% in men, enhancing the public awareness of the global burden of this chronic condition [91].

The correlation between airways type-2 related disorders and OSA has been further recently investigated in a cohort of 601 subjects enrolled in the World Trade Center Health Program. This analysis showed that after the disaster on 11 September 2001, patients who had developed or worsened CRS, possibly due to inhalation of environmental pollutants, presented a statistically significant higher risk for OSA with an odds-ratio of 1.80 (*p*-value = 0.006), after adjustment for possible confounding factors [92]. A further confirmation has been recently provided by a multicentric Spanish study, enrolling asthmatic patients who were synchronously diagnosed of moderate to severe OSA. In this analysis, continuous positive airway pressure (c-PAP) treatment demonstrated to improve both lung function and asthma control, with a reduction of patients with uncontrolled asthma from 41.4% to 17.2% after 6 months of treatment [93].

Obstructive sleep apneas have many different severe implications on the “extra-airways” district, due to several pathogenic mechanisms. Despite apneic episodes traditionally being associated with increased carbon dioxide (CO_2_) level, oral breathing-induced tachypnea may result in both central vasoconstriction and in decreasing the respiratory drive, promoting further central apneic events. Moreover, by means of “Bohr effect”, a decrease in CO_2_ plasmatic concentration eventually turns in a further reduction in tissue oxygen release, resulting in intermittent hypoxia, high sympathetic nervous activity, hypertension, endothelial dysfunction, oxidative stress, inflammation, and accelerated atherosclerosis, which have been investigated and confirmed in several population studies [94,95,96]. Finally, increased oxidative stress also implicates depletion in circulating electrolytes, which are often a cofactor of several antioxidant enzymes. In particular, according to a recent meta-analysis by Al Wadee et al., OSA patients showed significantly reduced serum magnesium levels, which correlated with worsening of cardiovascular risk biomarkers (e.g., C-reactive protein, ischemia-modified albumin, carotid intima-media thickness) [97]. Therefore, it is well established that the borders of this condition overcome the UA and that OSA, similarly to airways type-2 related disorders, should be addressed as a systemic syndrome.

### 3.2. Nitric Oxide: A Systemic Mediator

What is the role of systemic mediators in the pathogenesis of UA type-2 related disorders? Nitric monoxide (NO), for example, is an endogenous molecule of paramount relevance into several pathways, including blood vessel dilatation, myorelaxation, stimulation of hormone release and regulation of neurotransmission. Nitric monoxide production is catalyzed through the conversion of its precursors, L-arginine and oxygen, into citrulline by the enzyme nitric oxide synthase (NOS), which is displayed in different isoforms, including one constitutive and one induced (iNOS) form. The latter is responsible for the increase in NO levels in response to inflammatory stimuli, thus providing an innate defense mechanism due to NO’s free radical properties [98]. In asthmatics, increased NO levels contribute to hyperemia, edema, vasodilatation which, in susceptible airways, finally exacerbate bronchial narrowing. Moreover, recent research suggested that the inflammatory microenvironment results in upregulation of arginase activity, thus limiting L-arginine bioavailability for iNOS, eventually contributing to further free radical formation (e.g., peroxinitrite) with enhanced cytotoxic action in the airways [99]. Therefore, the measurement of NO has proven to be a reliable method for assessing airway inflammation. Various techniques, specifically tailored to different conditions, have been developed, including fractional exhaled NO (FeNO) and nasal NO (nNO). Fractional exhaled NO assessment is a non-invasive and easily administered tool, which has become a standard test in the diagnosis and treatment of asthma. The ERS/ATS has outlined a standardized procedure for this purpose [100]. An increased FeNO has also been demonstrated In type-2 related sinonasal diseases, enhancing the concept of the united airways. According to recent research by Kambara et al., significantly higher FeNO values were observed in non-asthmatic CRSwNP patients than in CRSsNP, whereas no significant difference in FeNO was observed between patients with CRSwNP with and without asthma [101]. Similarly, at a nasal level, a previous study by Baraldi et al. demonstrated that intranasal steroids significantly reduced nNO levels in children with perennial allergic rhinitis, thus speculating a possible role for nNO measurement in type-2 related nasal conditions [102]. From a physiological perspective, however, it should be considered that the higher amount of iNOS expression and NO production is located in the paranasal sinuses, rather than In the rest of nasal mucosa. According to the current literature, indeed, sinonasal obstruction as it develops in CRSwNP is responsible for a decrease in nNO measurement compared to both CRSsNP patients and healthy controls, mainly as a result of the ostiomeatal blockage by polyps, which reduces NO passage from sinuses to the nasal lumen, despite its increased synthesis due to inflammation [103].

Unlike FeNO, nNO is not routinely measured in rhinology clinical practice, mainly due to scarcity of specific tools as well as standardized techniques. Nevertheless, nNO normalization has been demonstrated after both medical and surgical opening of sinonasal ostia, thus providing evidence of its possible application as a marker of treatment response [104]. Moreover, Paoletti et al. demonstrated the effectiveness of anti-IL-4 receptor alpha monoclonal antibody (dupilumab), proving a rapid improvement in nNO values and other markers of disease severity as early as 15 days after the initiation of treatment [105].

Whenever it exceeds the physiological concentrations, NO has been unambiguously proposed as a marker of systemic inflammation [106]. Therefore, the evidence of its altered metabolism in most of type-2 related airways diseases further enhances the necessity for a multidisciplinary approach to address a pathology with implications that are progressively less confined exclusively to the nasal cavity.

## 4. A Syndemic Condition

The possible implications of a dysregulated type-2 inflammatory pattern overcome the boundaries of an organic and systemic setting.

A modern approach to CRSwNP should overtake the knowledge of the etiopathogenetic mechanisms, considering the relevant burden that this condition bears on the individual sphere, which inevitably brings huge implications on society (Figure 1).

### 4.1. Burden

#### 4.1.1. Epidemiological

In Western countries, interest in type-2 related upper airway diseases has been raising recently due to the awareness of its increasing epidemiologic trend and the comprehension of its socioeconomic burden. The worldwide prevalence of allergic rhinitis is probably underestimated because it is not considered a life-threatening condition, even though it has been established that it can lead to significant morbidity [107]. Moreover, prevalence studies have employed various definitions and methodologies, resulting in a broad range of reported outcomes. A recent multicentric, cross-sectional, epidemiological study (known as “SNAPSHOT program) was conducted on a random sample of 33.486 Middle Eastern adults who were screened for allergic rhinitis using the Score for Allergic Rhinitis questionnaire. Overall, the adjusted prevalence of allergic rhinitis in the adult general population in these countries was 5.6%, ranging from 3.6% in Egypt, to 6.4% in Turkey [107]. In accordance with the united airway perspective, allergic asthma was the most strongly related co-morbidity (odds ratio: 6.6). In high-income countries, allergic rhinitis seems to be more prevalent, with documented prevalence rates varying from 10 to 23.2% across different age brackets in the United States and Europe [108]. While pinpointing the exact incidence remains challenging, existing evidence indicates that, akin to other type-2 airway-related disorders, the prevalence of medically diagnosed allergic rhinitis has exhibited an upward trajectory over time. According to a recent metanalysis by Licari et al., the prevalence of allergic rhinitis in pediatric population raised from 8.39% in the period 2012–2015 to 19.87% in the period 2016–2022, possibly due to both extrinsic (e.g., microenvironmental) and intrinsic (e.g., type-2 dysregulated immune response) factors [109].

The first European International multicentric study to assess the prevalence of CRS was performed by the GA^2^LEN network of excellence, which collected the answers of a random sample of 57,128 adults aged 15–75 years in 19 different centers in Europe. According to this wide survey, the prevalence of CRS was estimated to be 10.9%, with marked geographical variation varying from 6.9% (Germany) to 27.1% (Portugal) [110]. Similarly, data from a cross-sectional investigation that randomly included 10,636 respondents from seven Chinese cities found an overall mean prevalence of 8.0% (range 4.8–9.7%) [111]. Nevertheless, prevalence estimates for CRS obtained from population-based survey responses may have inappropriately increased the true prevalence rate, since current evidence suggests that approximately 40 to 50% of “self-reported” CRS lacks objective evidence of inflammation using CT [112]. In fact, in a recent study enrolling a randomly selected group of 834 patients, who referred to an European radiology department for imaging of the head with non-rhinologic indications, the prevalence of clinically based CRS, according to the EPOS guidelines, was found to be up to 6.4% [113].

CRS is particularly prevalent among people with specific medical conditions, including allergic rhinitis, chronic obstructive pulmonary disease, and gout and up to 38% in patients with severe asthma, particularly those with late-onset eosinophilic severe asthma.

To overcome any selection bias and to provide reliability to the collected data, the worldwide prevalence of asthma, allergic rhinoconjunctivitis, and AD was assessed with a standardized methodology by the International Study of Asthma and Allergies in Childhood (ISAAC) Steering Committee. Overall, 463,801 children aged 13–14 years from 155 centers in 56 different countries were included. The most striking evidence provided by this analysis was the definite confirmation of the huge variation in the prevalence of type-2 related conditions, varying from 20-fold to 60-fold between different countries worldwide. The highest prevalence was reported in higher-income countries (e.g., UK, Australia, New Zealand, and Republic of Ireland), whereas lower rates were observed in lower-income states [114]. Over the past two years, the International Study of Asthma and Allergies in Childhood (ISAAC) data has been revisited using the same core methodology by the Global Asthma Network (GAN). The GAN has released updated prevalence and time trend information for asthma, now extending its scope to include adults for the first time. This comprehensive study involved multiple centers, extrapolating data from 193,912 adults across 17 countries. The reported overall prevalence rates were 4.4% for asthma and 9.9% for atopic dermatitis (AD), with significant variations observed both between and within countries (ranging from 0.9% to 29.0% for asthma and 1.6% to 29.5% for AD, respectively) [115]. As concerning the younger population, results were slightly higher, with 10% of adolescents (13–14 years of age) and children (6–7 years of age) experiencing asthma symptoms ever, and 11% of adolescents and 13% of children having reported AD. Again, a significant part of the whole variability was attributable to the economic income of each analyzed country [116].

In further support of the pathophysiological basis that connects type-2 related diseases, the prevalence of CRS and asthma has been linked, with up to 50% asthma prevalence in CRSwNP patients [117]. Data from the UK National Chronic Rhinosinusitis Epidemiology Study showed that CRS patients were more likely to suffer from asthma and aspirin sensitivity, with those with NPs subtypes even more likely to report asthma including the majority of those with AFRS. In this cohort composed by 1470 participants, prevalence of asthma ranged from 21% in CRSsNP through 47% in CRSwNP to 73% in AFRS [118].

The epidemiological relationship between CRS and AD have been recently addressed in a large population-based, longitudinal study including 5638 CRS patients and 11,276 non-CRS as a control group from the Korean National Health Insurance Service database. Results from this study showed that patients with CRS were at higher risk of developing AD, having shown an incidence rate for AD of 63.59 per 1000 person years in the CRS group and 45.38 per 1000 person years in the comparison group. As expected, no increased risk was observed in the CRS group for developing non-type-2 related dermatologic conditions, such as vitiligo or psoriasis, compared to patients without CRS [119].

Finally, an observational study by Khan et al. assessed the prevalence of co-existing type 2 inflammatory conditions. At least one type-2 related disease was identified in 66% of asthmatics, 69% of CRSwNP patients, and 46% of those suffering from AD. Moreover, 24%, 36%, and 16% of patients suffering from asthma, CRSwNP and AD had at least two type-2 related disorders, respectively [120].

#### 4.1.2. Economic

The most immediate and impactful consequence of the high epidemiological burden is the substantial economic cost associated with CRS and type-2 related diseases for the whole society. Direct healthcare expenses, such as medication, hospitalization, physician examinations, and surgery, represent only a portion of the costs. Simultaneously, there is a significant and comparable indirect cost arising from absenteeism, disability, resulting in a loss of productivity and work performance. According to a recent review by Rudmik et al., the overall direct cost of CRS is estimated to range between $10 and $13 billion per year (e.g., 10.000–13.000 million dollars/year) in the USA, while the overall indirect cost due to CRS-related losses in work productivity is estimated to be in excess of $20 billion per year, for a total of more than $30 billion/year [121]. In another study conducted in the English population, CRS subjects were shown to claim a significantly higher total number of primary care visits than the control group (per year: 4.14 vs. 1.16, *p* < 0.001). On the utilization of secondary care services, CRS subjects recorded a higher outpatient interaction (2.61 vs. 0.40, *p* < 0.001) with a mean total cost of £613.58, as compared to £97.40 in the control group. Concerning the indirect costs of CRS, the average total missed workdays in CRS patients were estimated annually at 18.7 per patient [122].

Equally dramatic is the economic burden of the sleep-related breathing disorders possibly associated with upper-airways type-2 conditions. A recent systematic literature review promoted by Bocconi University (Milan, Italy) assessed the economic burden of OSA in the adult Italian population considering the costs of possible systemic OSA-related implications (e.g., hypertension), the costs due to OSA diagnosis and treatment, and the economic value in terms of a worsened Quality of Life (QoL) due to OSA undertreatment. From this analysis emerged an economic burden ranging from EUR 10.7 to EUR 32.0 billion/year in Italy. The cost of impaired QoL was calculated between EUR 2.8 and EUR 9.0 billion/year. Moreover, the authors emphasized that these figures were notably higher than the current costs incurred for the diagnosis and treatment of obstructive sleep apnea (OSA), which amount to EUR 234 million per year [123].

Recent studies suggested that the socioeconomic status, in relation to the daily habits and behaviour, may have implications on the different microenvironmental stimuli patients are exposed to, eventually turning into a decisive factor within the complex clinical scenario. In recent research, a lower income predicted higher exposure to air pollution, in terms of PM_2_._5_, black carbon, and nitrogen dioxide, which eventually resulted in higher CRS severity as measured by an increased need for chronic OCS treatment [124].

In USA, data from the National Hospital Ambulatory Medical Care Survey (NHAMCS) found that, annually, 20% of asthmatics, from those who need emergency service visits, have at least one hospital admission with an average of 3.6 days of hospitalization [125]. Overall, in a European setting, estimated mean annual asthma-related costs per patient with severe asthma amounted to EUR 6500, of which approximately EUR 2400 and EUR 4100 were direct and indirect costs, respectively. Furthermore, higher direct costs were proven in patients regularly on treatment with OCS compared to those who were under different medications [126].

Similarly, social costs of moderate-to-severe forms of AD across Europe have been estimated at EUR 30 billion per year; EUR 15.2 billion related to missed workdays or reduced work productivity, EUR 10.1 billion related to direct medical costs and EUR 4.7 billion related to personal expenditure of patients/families [127]. In a population-based, controlled cohort study in Sweden, the annual mean per-patient direct healthcare costs in the first year following diagnosis of AD were EUR 941 and EUR 1259 higher in pediatric patients with mild-to-moderate and severe AD, respectively, compared to controls. Unadjusted annual health care costs in 2018 were $4979 higher for adults with atopic dermatitis ($14,603) than for the matched controls ($9624), driven by outpatient services and pharmacy [128].

Current evidence depicts the dramatic burden of type-2 related diseases in the global economic scenario, assuming the features of a syndemic condition. Enhancing the awareness of the potential submerged expenses due to undiagnosed or wrongly treated patients, it will improve the efficiency of the whole healthcare process in for the treatment of patients affected by type-2 related conditions.

#### 4.1.3. Social

As we have said before, a new personalized strategy has begun to gain traction, considering not only the broad spectrum of pathophysiological mechanisms, but also the individual differences in people’s genes, environments, and lifestyles. It fits well in this new approach the paradigm of the “4-P Medicine” (Prediction, Prevention, Personalization, Participation), introduced by the molecular biologist and oncologist Leroy Hood, which is especially suited for patients affected by chronic conditions such as airways type-2 related disorders [129]. From this perspective, patient’s participation in the treatment plan is acquiring an increasingly pivotal role and a doctor-patient counselling to maintain adherence and compliance to treatment has become part of the healthcare process.

In the whole scenario, since response to treatment is influenced by several factors, patients’ stratification is fundamental to setting the correct diagnostic and therapeutic path. Several PROs have been proposed ad validated in the literature for CRS, in order to objectively quantify the entity of pre-existing or residual symptoms from a patient-centered perspective. Since PROs provide patients’ own interpretation of their health status without modification by a clinician, in recent years several randomized controlled trials and real-life experiences have made use of these tools, including disease-specific questionnaires (e.g., Sino-Nasal Outcome Test-22) and generic ones (e.g., Short Form-36), as reliable endpoints for new drugs [130,131].

Quality of life is influenced by many factors, related to both the sociodemographic characteristics of the patients and possible comorbid conditions. CRS patients can experience a complex range of discomforts beyond those simply related to the nose and paranasal sinuses. Indeed, even though CRS is marked by sinonasal symptoms, the most severe QoL complains have been demonstrated those affecting general-health-related domains. Diminished sleep, productivity, cognition, mood, and fatigue are associated with the decision to elect surgical intervention as well as with a poorer QoL [132].

According to a systematic review by Worth et al., PROs found to be sufficiently well validated to offer promise for use in clinical settings for the assessment of QoL in asthmatic patients were the Asthma Quality of Life Questionnaire (AQLQ) and mini-AQLQ for adults, and Pediatric Asthma Quality of Life Questionnaire for children [133]. Using these tools, patients exhibiting better symptom control have been demonstrated as referring a generally lower social burden of disease, since AQLQ scores tended to decrease significantly as the frequency of asthma exacerbations increased. The Quality of Life scores in the “Emotional function” and “Environmental stimuli” subscale of the AQLQ decreased significantly as time from onset increased, thus emphasizing the importance of early diagnosis and management of asthma related symptoms to stop the progression of disease [134].

Furthermore, sleep non-breathing-related disorders have been associated with asthma. Insomnia (defined by Insomnia Severity Index score ≥ 10) was identified in 263 participants (37%) with physician-confirmed asthma enrolled in the Severe Asthma Research Program. Presence of insomnia was associated with higher levels of depression, anxiety symptoms and poorer quality of life, as well as a 2.4-fold increased risk of having a not well-controlled asthma and a 1.5-fold increased risk for asthma-related health care utilization compared to patients without insomnia [135]. Similarly, Leander et al. demonstrated a statistical correlation between anxiety, depression, and asthmatic symptoms, including attacks of shortness of breath after activity, impairing the QoL [136].

Quality of life assessing tools for AD are divided into self-reported (for older children and adolescents) or proxy-reported (for infants and younger children) depending on the ages of the respondents, both further classified into skin-specific and disease-specific questionnaires. One of the most commonly used skin-specific tools to assess QoL in AD patients is the Dermatology Life Quality Index in adults and related questionnaires for children, such as the Children’s Dermatology Life Quality Index [137]. Health-related QoL has been demonstrated often as markedly impaired in both adults and children with AD. In a recent metanalysis, AD adults showed a significantly lower QoL compared to controls and disease severity showed positive correlation with a poorer QoL. Nevertheless, authors noticed high heterogeneity in the tools used to measure the disease severity across different studies, therefore further validation analysis is warranted in order to cross-culturally validate definite reliable PROs in patients with AD [138].

## 5. Conclusions

Type-2 related disorders are a heterogeneous disease, yet they share a common pathophysiological mechanism and primarily affect the united airways. The definite clinical expression (phenotype) of CRS, bronchial asthma, AD and other type-2-related manifestations is the result of a complex interaction between the internal susceptibility (genetic predisposition, severity of UA type-2 inflammation, microbiome, together with the presence of relevant comorbidities) and external determinants (environmental and behavioral factors); therefore, an “endotype-based” classification is mandatory in clinical practice, aimed to offer a tailored personalized approach.

In this article, we particularly focused on airways type-2 related disorders from both a pathophysiological objective and a psychosocial subjective point of view, as they were demonstrated to affect the whole dimensions of wellness: emotional, occupational, social, spiritual, intellectual, environmental, financial, and physical. In light of the relevant burden on both the epidemiological, economic, and social sphere, we aimed to promote a novel perspective on airways type-2 related disorders, beyond the limits of a locally bordered condition, but through the concept of a syndemic disease.

## Figures and Tables

**Figure 1 ijms-25-00730-f001:**
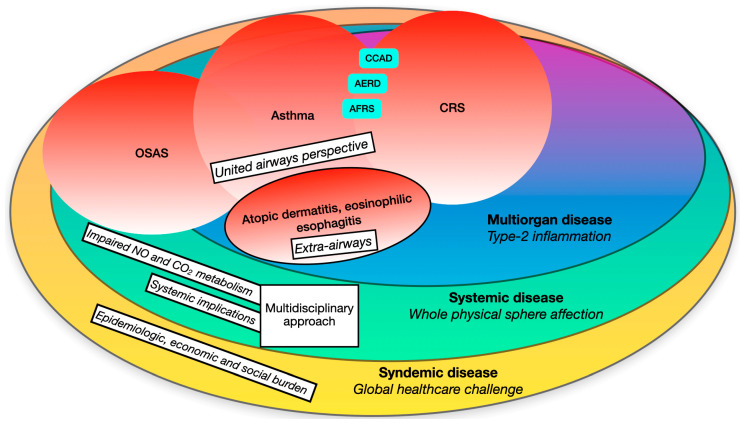
Broad view on type-2 related disorders from a multiorgan, systemic and syndemic perspective.

## Data Availability

Not applicable.

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
