# Peer review of "Airways Type-2 Related Disorders: Multiorgan, Systemic or Syndemic Disease?"

_ijms, 2024, doi:10.3390/ijms25020730_

Round 1
Reviewer 1 Report
Comments and Suggestions for Authors
The paper” Airways type-2 related disorders. Multiorgan, systemic or syn-2 demic disease?” by Giombi et al. is an interesting paper that can impact how we think about these diseases.
The following changes hould be done.
1. You should provide some epidemiological data on the prevalence of rhinitis, by data I mean some numbers. In the same way, when it is indicated that it has increased, numbers should be provided.
2. Authors should mention big epidemiological studies such as ISAAC (International Study of Asthma and Allergies in Childhood) and GAM (Global Asthma Network)
3. Stratcham hypothesis has been criticized. There are authors in favor and authors agaisnt . The authos should include this references and discuss its r criticism in their review.
https://www.frontiersin.org/articles/10.3389/fimmu.2021.635935/full
https://www.frontiersin.org/articles/10.3389/fimmu.2021.635522/full
https://www.pnas.org/doi/pdf/10.1073/pnas.1700688114
4. When speaking for first time of billion (line 519) indicate if you mean 1,000 millions (10^9) or 1,000,000 millions (10^12). (There is a difference between the meaning of this term in the United States and Europe) See https://en.wikipedia.org/wiki/Billion#:~:text=Billion%20is%20a%20word%20for,defined%20on%20the%20short%20scale.
5. In the bibliography When a web is mentioned for example
109. National Hospital Ambulatory Medical Care Survey (NHAMCS), available at 894 http://www.cdc.gov/nchs/data/ahcd/nhamcs_emergency/2010.pdf
The date that was accessed should be included.
6 The obsolescence index (Median of the bibliographic references is six years) , try to update the paper with more updated bibliography.
Reviewer 2 Report
Comments and Suggestions for Authors
The review article by Giombi et.al focuses on the paradigm shift in understanding Chronic Rhinosinusitis (CRS) and its relationship to type-2 inflammation. It discusses how type-2 related disorders, including asthma and atopic dermatitis, extend beyond the respiratory system, affecting multiple organs and requiring a multidisciplinary approach. The review emphasizes the systemic and syndemic nature of these conditions, considering their increasing epidemiological, economic, and social burden. It advocates for a comprehensive perspective on these disorders, highlighting their implications in various aspects of health and disease. The review is written in good logic. I have some minor comments for the author’s consideration.
1. The author may need to highlight that although the disease like asthma and atopic dermatitis can be categorized as type-2 inflammation, by sharing the same inflammatory cytokines, like IL-4, IL-5, and IL-13. Inflammatory cytokines are indeed significant in amplifying inflammation in these diseases, but they are not necessarily the primary cause. It's crucial to acknowledge that each disease has unique pathophysiological features and triggers. Asthma primarily affects the airways, with factors like airway hyperreactivity playing a significant role, while AD is a skin condition with a strong component of barrier dysfunction.
2. Due to the distinct mechanisms within the type-2 inflammation, the clinical therapy considerations may be different and can be highlighted in the review. For example, asthma is caused by the imbalance of goblet cells and ciliated cells, thereby the reconstitution of MCC function (i.e. Jag-Notch2 blockage) may be the primary therapy consideration, and the inhibition of cytokines as auxiliary therapy. For AD, the reconstitution of skin barrier function should be the primary therapy consideration (like SPINK5, KLK5/7 inhibition, etc).
Comments on the Quality of English Languagenone
